# Additive Manufacturing and Physicomechanical Characteristics of PEGDA Hydrogels: Recent Advances and Perspective for Tissue Engineering

**DOI:** 10.3390/polym15102341

**Published:** 2023-05-17

**Authors:** Mohammad Hakim Khalili, Rujing Zhang, Sandra Wilson, Saurav Goel, Susan A. Impey, Adrianus Indrat Aria

**Affiliations:** 1Surface Engineering and Precision Centre, School of Aerospace, Transport and Manufacturing, Cranfield University, Bedford MK43 0AL, UK; m.hakim-khalili@cranfield.ac.uk (M.H.K.); s.a.impey@cranfield.ac.uk (S.A.I.); 2Sophion Bioscience A/S, Baltorpvej 154, 2750 Copenhagen, Denmark; ruz@sophion.com (R.Z.); swi@sophion.com (S.W.); 3School of Engineering, London South Bank University, 103 Borough Road, London SE1 0AA, UK; goels@lsbu.ac.uk; 4Department of Mechanical Engineering, University of Petroleum and Energy Studies, Dehradun 248007, India

**Keywords:** hydrogels, poly(ethylene glycol)diacrylate, tissue engineering, three-dimensional, printing

## Abstract

In this brief review, we discuss the recent advancements in using poly(ethylene glycol) diacrylate (PEGDA) hydrogels for tissue engineering applications. PEGDA hydrogels are highly attractive in biomedical and biotechnology fields due to their soft and hydrated properties that can replicate living tissues. These hydrogels can be manipulated using light, heat, and cross-linkers to achieve desirable functionalities. Unlike previous reviews that focused solely on material design and fabrication of bioactive hydrogels and their cell viability and interactions with the extracellular matrix (ECM), we compare the traditional bulk photo-crosslinking method with the latest three-dimensional (3D) printing of PEGDA hydrogels. We present detailed evidence combining the physical, chemical, bulk, and localized mechanical characteristics, including their composition, fabrication methods, experimental conditions, and reported mechanical properties of bulk and 3D printed PEGDA hydrogels. Furthermore, we highlight the current state of biomedical applications of 3D PEGDA hydrogels in tissue engineering and organ-on-chip devices over the last 20 years. Finally, we delve into the current obstacles and future possibilities in the field of engineering 3D layer-by-layer (LbL) PEGDA hydrogels for tissue engineering and organ-on-chip devices.

## 1. Introduction

Hydrogels have been extensively studied in recent years as potential solutions for biomedical applications such as tissue engineering and organ-on-chip devices. This is due to their biocompatibility and ability to be tailored for specific physical, chemical, and mechanical properties [1,2,3,4,5]. Additionally, the structure and properties of hydrogels mimic the micro-environment of various human organ tissues [6,7,8,9,10,11,12,13] and thus, hydrogels are widely used in a diverse range of tissue engineering and organ-on-chip devices [14,15,16]. Poly(ethylene glycol) diacrylate (PEGDA) hydrogels have garnered significant interest among hydrogel materials due to their remarkable properties, including hydrophilicity. As a result, they have been extensively explored and applied in various areas, such as three-dimensional (3D) tissue-engineered constructs, bio-sensing mediums, and drug-controlled release matrices [17,18,19,20].

Due to its biocompatibility and ability to mimic the extracellular matrix (ECM) of living tissues, PEGDA hydrogel has found extensive use in various biomedical applications [21,22]. PEGDA hydrogels are synthesized
by the cross-linking of PEG diacrylate monomers, which results in a three-dimensional network of polymer chains that can retain large amounts of water [23]. The first reported use of PEGDA hydrogel in biomedical applications was in the 1970s, when it was used as a scaffold for in vitro cell culture. PEGDA hydrogels were found to be biocompatible and non-toxic, making them suitable for use in cell culture studies (Figure 1) [24,25]. In the 1980s, PEGDA hydrogels were further developed for use in drug delivery applications. The hydrogels were found to be able to retain drugs in their pores and release them in a controlled manner over time [26]. This property made PEGDA hydrogels suitable for use in the sustained release of drugs for the treatment of chronic diseases [27]. In the 1990s, PEGDA hydrogels were also used in injectable wound healing applications [28]. These hydrogels were found to be able to promote the growth of new blood vessels, and to promote the growth of new tissue in wounds. In the 2000s, PEGDA hydrogels were further developed for use in tissue engineering applications [29,30]. The compatibility of hydrogels with diverse cell types, including stem cells, has been demonstrated, and they facilitate the development of new tissue [31,32]. The capacity of PEGDA hydrogels to imitate the extracellular matrix (ECM) of living tissue enhances their appeal for tissue engineering applications [33,34,35,36,37,38]. This property made PEGDA hydrogels suitable for use in the regeneration of various types of tissues, such as cartilage and bone [39,40,41,42]. In recent years, PEGDA hydrogels have been further developed for use beyond tissue engineering into other biomedical applications, such as in vivo drug delivery and wound healing [43,44,45,46]. The hydrogels have also been used in various other applications, such as in the fabrication of micro- and nano-devices [47,48,49].

**Figure 1 polymers-15-02341-f001:**
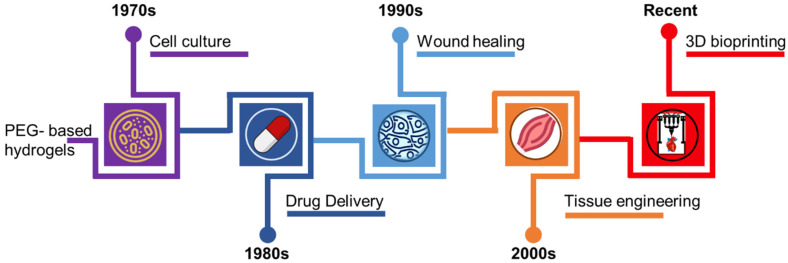
An overview of use of PEG-based hydrogels in biomedical and biological applications.

Moving away from molding and bulk cross-linking processes, 3D printing using stereolithography (SLA) and digital image processing (DLP) among other advanced printing techniques offer a faster and more dependable manufacturing approach for creating complex scaffold shapes with reliable mechanical properties [50,51,52,53]. The use of 3D printed PEGDA hydrogels has been studied extensively for developing skeletal muscle microtissues and contractile cardiac tissue as the demand for PEGDA hydrogels continues to grow (Figure 2). However, there are still critical gaps in knowledge regarding the physicochemical and mechanical behavior and properties of 3D printed PEGDA hydrogels. Previous papers mainly focused on the synthesis of the PEGDA hydrogels and material design and fabrication of bioactive hydrogels and their cell viability and interactions with the ECM and neglected the importance of studying the fabrication processes and how different parameters can influence the physicochemical and mechanical properties of the 3D PEGDA hydrogels [54,55]. The impact of layer-by-layer (LbL) fabrication on number of intrinsic properties of photo-cross-linked 3D printed PEGDA hydrogels including their chemical, mechanical, and dimensional stability, physical isotropy behavior, and degree of cross-linking has to be studied in further details [56]. Moreover, more investigation is required to determine the basic nanomechanical behavior of 3D PEGDA hydrogels, and how different fabrication parameters such as layer thickness and ultraviolet (UV) dosage can affect their surface properties and interfacial layer-layer adhesion and homogeneity in mechanical properties across the entirety of the 3D printed structures [57]. Methods such as nanoindentation and atomic force microscopy (AFM) are well-suited for examining surface characteristics, including the spatial distribution of mechanical properties, which play a vital role in cell adhesion, tissue formation, and evaluating the alignment between layers in multilayer 3D printed structures created using layer-by-layer (LbL) techniques [58,59,60,61]. The nanomechanical tests can also be used as an indirect but more accurate method for evaluating the degree of cross-linking in a 3D printed structures and map the variations in mechanical properties and infer the degree of cross-linking accordingly, which has proven to be very challenging to measure with advanced techniques such as nuclear magnetic resonance (NMR), small angle x-ray scattering (SAXS), and rheology [61]. Extensive research literature consistently highlights the significant impact of mechanical cues at the substrate interface, where cells are cultured, on various cellular behaviors. These behaviors encompass crucial aspects such as cell growth, migration, proliferation, differentiation, and tissue formation [62,63,64,65,66,67]. For example, the surface elastic modulus, topography, and adhesion of hydrogel substrates can direct the fate of stem cells which was reported on C2C12 skeletal myoblasts [68,69,70,71]. Moreover, the stiffness and surface topography of the ECM have a direct impact on elongation direction and sarcomere alignment of muscle tissues [68,69,70,71].

Addressing these gaps will allow for optimal use of these materials in various biomedical applications, including tissue engineering, drug delivery, and cancer therapy.

**Figure 2 polymers-15-02341-f002:**
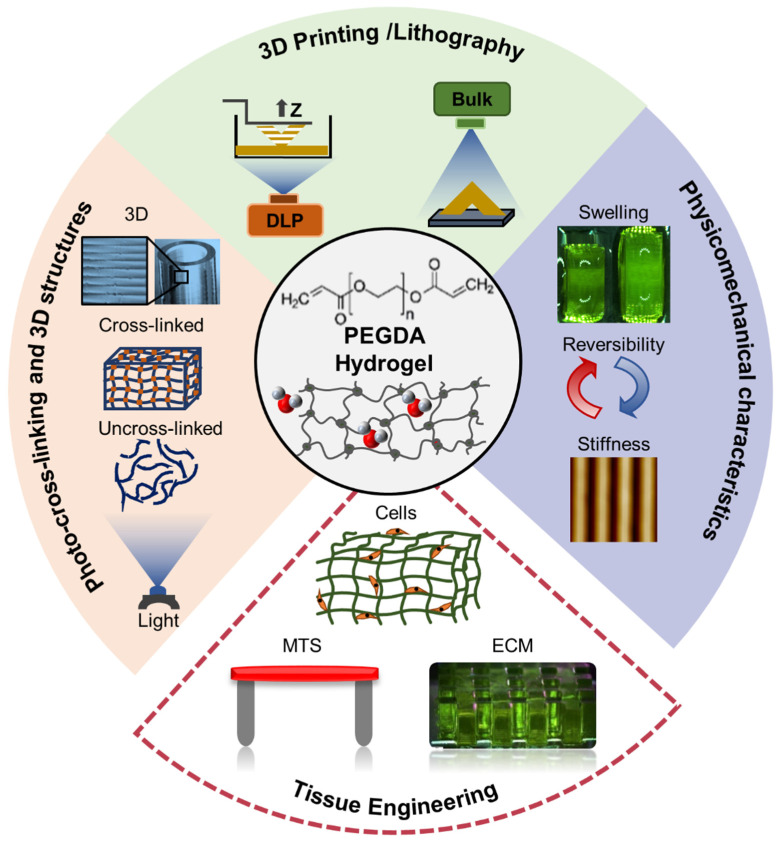
A schematic showcases PEGDA hydrogel in the field of tissue engineering, covering material cross-linking, fabrication techniques, physicomechanical characteristics, and clinical applications.

## 2. Manufacture of 3D PEGDA Biostructures

Scalability is a vital aspect of 3D printing for the rapid production of 3D biomimetic structures in tissue engineering and regenerative medicine. This technology enables precise control over structure and size, customizable to specific requirements, while also facilitating the cost-effective production of large quantities of intricate structures. Therefore, the combination of additive manufacturing techniques, biomaterials, cells, and the scalability of 3D printing offers a powerful approach for the development of functional tissues and organs that can meet the clinical demands of tissue engineering and regenerative medicine. Various 3D bioprinting strategies have been developed over the past decade, each with its own unique advantages and limitations. In this section, we will discuss their working principles, and their suitability for different applications (Figure 3a).

### 2.1. 3D Bioprinting Techniques

Extrusion-based 3D printing is a technique which involves creating structures made from melted materials (filament) using rollers and a heating system. However, the technique has undergone rapid development and expansion into bioprinting, becoming the most popular technique in the field. Filaments are replaced with hydrogels and bioinks containing living organisms, which are extruded through a shear-thinning process rather than being melted through the application of heat [72,73]. Extrusion-based bioprinting utilizes pneumatic or mechanical fluid dispensing systems, with pneumatic systems being valve-free or valve-based, and mechanical systems driven by a piston or screw (Figure 3b). Extrusion-based bioprinting offers multiple benefits, including the ability to deposit high-viscosity bioinks and achieve high cell densities. Moreover, it allows for uninterrupted continuous extrusion of bioinks [74,75]. However, their bioprinting speed is relatively slow and cell viability is usually moderate due to high shear stresses experienced by cells during the extrusion process [76,77]. Despite these limitations, extrusion-based bioprining has been widely adopted in bioprinting due to its simplicity in instrumentation [78]. In an experiment, a combination of PEGDA hydrogel and charged monomers such as 3-sulfopropyl acrylate, along with Poloxamer 407 (P407), were used for extrusion printing. The material was successfully printed in a pyramidal shape but its use for direct cell encapsulation might be limited due to P407 cytotoxicity [46]. Another study used it to produce human-scale constructs such as nose and ear using a combination of gellan gum and PEGDA. Murine bone marrow stromal cells (BMSCs) and a mouse osteoblastic cell line (MC3T3-E1) were encapsulated in the constructs, which showed structural stability, biocompatibility, and non-toxicity [79]. The technique was further developed to create 3D printed tracheal utilizing PEGDA, sodium alginate, and calcium sulphate using a home-made extrusion bioprinter with assisted UV laser. The results showed that by optimizing the printing parameters such as laser power intensity the stiffness of the printed biological tissue can be altered to match that of the biological tissue or organ [80]. Extrusion based printing technologies are developed further to include multi-functional, multi-material 3D printing that can be utilized to print PEGDA hydrogels with fabrication strategies for living tissues and soft robotics [81]. For example, a mixture of gellan gum and PEGDA hydrogels with encapsulated BMCs and MC3T3-E1 cell were successfully printed using extrusion 3D printing and they showed a cell viability of over 87% [79].

Inkjet printing enables the precise and non-contact deposition of small quantities of materials on diverse surfaces. It offers two types of printing: continuous and drop-on-demand (DOD) [82], in which cells and biomaterials can be patterned into desired substrates by repeatedly depositing droplets at pre-designed locations to form a structure (Figure 3c) [83]. Inkjet bioprinting provides higher resolution but may have limitations in terms of achievable cell densities and printing speed, as compared to extrusion-based bioprinting [84]. A biocompatible ink was created by combining Pluronic F127 (PEO-PPO) and PEGDA. The gel structures obtained remained stable when exposed to an aqueous environment and were successfully seeded with fibroblast cells using an inkjet printer as the delivery method [85]. Another biocompatible ink was developed by synthesizing a photo-cross-linkable polycaprolactone dimethylacrylate (PCLDMA) and mixing it with PEGDA. They optimized the ink properties under a nitrogen atmosphere during printing, demonstrating its potential for 3D structure production with good biocompatibility [86]. In another study, PEGDA hydrogel and Irgacure 2959 photoinitiator were combined to create micropatterns on the surface of a thin PEGDA layer that was deposited on a polycarbonate substrate using inkjet printing. The micropatterns (wrinkles) were created through plasma etching and were intended to enhance cell adhesion and differentiation [87].

The manufacture of PEGDA hydrogels using microfluidics is a cutting-edge technique that utilizes microfluidics channels to produce hydrogels with a high degree of control over their properties (Figure 3d) [88]. Microfluidics is a field of research that deals with the manipulation of fluids at a microscale, and microfluidic devices are designed to control the flow of fluids through tiny channels. In the case of PEGDA hydrogels, microfluidic devices are used to control the flow of the monomers and initiators that are used to create the hydrogel [89]. This allows for precise control over the concentration and ratio of the monomers and initiators, which in turn allows for precise control over the properties of the resulting hydrogel [90]. For example, by controlling the flow of the initiator, it is possible to achieve different degrees of cross-linking in the hydrogel, which will affect the mechanical properties of the hydrogel [91]. One of the major advantages of microfluidic fabrication is the ability to produce PEGDA hydrogels with highly defined microstructures. For example, by using microfluidic devices, it is possible to create hydrogels as microspheres and micromolds [92].These microstructures are particularly useful in cell encapsulation and drug delivery applications, as they can be used to create hydrogels that mimic the natural microenvironment of cells and allow for controlled release of drugs [93,94,95,96].

Another way to create 3D PEGDA scaffold architectures is through the use of SLA and DLP which are vat photo-cross-linking technologies that share similarities but also exhibit significant differences (Figure 3e) [97,98,99,100,101]. In SLA, a vat of photopolymer resin is subjected to an ultraviolet (UV) laser. Using a UV laser, a predetermined shape is projected onto a photopolymer vat. The photopolymer reacts to UV light, creating a single layer of the intended 3D object. The build platform gradually lowers for each layer, repeating the process until the complete 3D object is formed [102]. DLP is a variation of SLA that utilizes a digital projector to flash a single image of the layer across the entire resin at once using UV or visible light [103]. DLP utilizes a digital micromirror device (DMD) chip composed of reflective aluminum micromirrors. These micromirrors redirect UV light from a projector to project a designed pattern or a layer of a 3D CAD model onto the resin [104]. Unlike SLA, which uses a UV laser beam, DLP maintains a stationary UV light source that cures the entire resin layer at once. This results in faster printing speeds but with a fixed resolution typically ranging from 20 to 100 microns [105]. On the other hand, SLA’s laser beam moves from point to point, allowing for higher accuracy and better-quality prints with intricate details with smaller than 25 micron XY resolution [106,107]. Additionally, SLA 3D printers have a more scalable build volume, whereas DLP printers are optimized for specific use cases. These two techniques and their application in creating 3D PEGDA hydrogels for tissue engineering are discussed further in the following sections.

**Figure 3 polymers-15-02341-f003:**
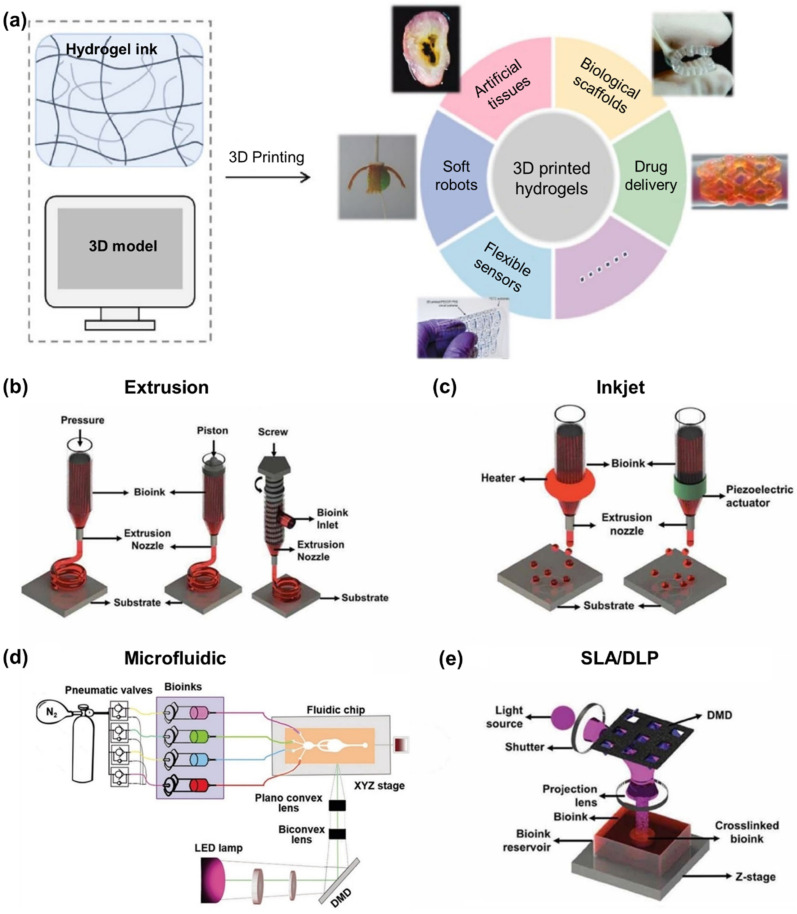
(**a**) The 3D printing of hydrogels for various biomedical applications [108]. Reproduced with permission from ref. [108]. Copyright 2022 Elsevier. Schematic of 4 different 3D printing techniques used for 3D PEGDA hydrogel fabrication, (**b**) extrusion [109], (**c**) inkjet [109], (**d**) microfluidic-based 3D printing [98], (**e**) SLA/DLP [109]. (**b**,**c**,**e**) Reproduced with permission from ref. [109]. Copyright 2019 Wiley. (**d**) Reproduced with permission from ref. [98]. Copyright 2018 Wiley.

### 2.2. Photo-Cross-Linking Mechanism

PEG, or polyethylene glycol, is a hydrophilic, water-soluble, and biocompatible polymer. In an aqueous solution, the PEG molecule is neutral, exhibits high mobility, and maintains hydration [110,111]. PEGDA is a derivative of PEG in which the hydroxyl groups at the ends of the PEG chains are replaced with acrylates. PEGDA hydrogels are considered synthetic hydrogels that are chemically cross-linked through the photo-cross-linking technique, in which a UV or visible light source is used to induce the cross-linking [112]. This rapid and easy synthesis technique allows for the creation of a soft polymer with tailored physical, chemical, and mechanical properties [113,114]. When PEGDA hydrogels are exposed to UV or visible light, the photoinitiator molecules absorb the light energy and generate free radicals. Free radicals react with the acrylate groups on the ends of PEG chains, forming covalent bonds between the chains (Figure 4). This results in the formation of a 3D cross-linked polymer network, which gives the hydrogel its mechanical strength and shape [115,116]. In brief, PEGDA hydrogels are created by dissolving PEGDA powder in a solvent at a specific ratio. A photoinitiator is then added, and the solution is exposed to UV light for a specific duration to initiate photo-cross-linking [117]. By altering the molecular weight, pre-polymer concentration, and photo-cross-linking conditions, the mechanical characteristics of the hydrogel can be modified. For instance, increasing the pre-polymer concentration can raise the hydrogel’s compressive modulus while simultaneously reducing its hydrolytic degradation rate [118].

**Figure 4 polymers-15-02341-f004:**
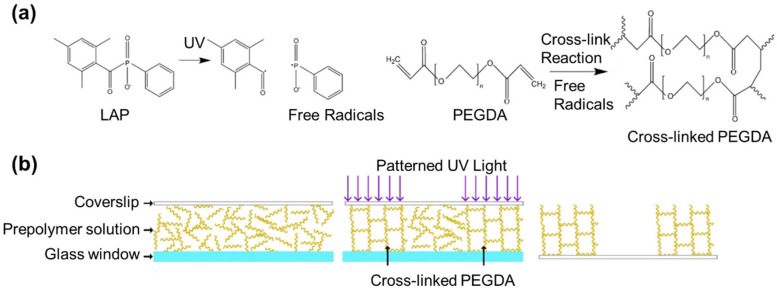
(**a**) A schematic illustrating the production of free radicals from a photoinitiator, lithium phenyl-2,4,6-trimethylbenzoylphosphinate (LAP), and the resulting cross-linking of PEGDA monomers by these free radicals. (**b**) A diagram depicting a typical photo-cross-linking process [119]. Reproduced with permission from ref. [119]; Copyright 2018 America Chemical Society.

### 2.3. Bulk vs. Layer-by-Layer Photo-Cross-Linking

Different techniques have been developed to perform photo-cross-linking on PEGDA, which include the utilization of bulk cross-linking, 3D printing, and microfluidic methods. Each of these techniques offers unique advantages and limitations and can be chosen based on the specific requirements of the application. The bulk cross-linking of PEGDA hydrogel is the most straightforward process among the other two techniques, it involves mixing the monomers and initiators together in a container and then cross-linking them under controlled conditions [120,121]. This method of cross-linking is known as “bulk” cross-linking because all of the monomers are added to the container at once and then cross-linked together (Figure 5a,b) [122]. The key to this process is controlling the conditions under which the monomers are cross-linked. This includes factors such as temperature, time, and the concentration of photoinitiator used. However, due to the nature of bulk cross-linking, it can lead to inhomogeneity in cross-linking, which can affect the mechanical properties of the hydrogel [123]. This inhomogeneity is caused by the distance of the light source to the hydrogel and the direction of light, which results in variations in the intensity of light reaching different parts of the hydrogel, leading to non-uniform cross-linking [124,125]. By carefully controlling these conditions and the distance and direction of the light source, it may be possible to minimize the inhomogeneity in cross-linking and mechanical properties, but it cannot be completely eliminated.

The process of 3D printing, or additive manufacturing, involves the creation of a three-dimensional object by incrementally adding layers of material. The layer-by-layer fabrication process is one of the most popular and widely used methods of 3D printing. In the layer-by-layer fabrication process, a 3D model of the object is sliced into thin layers. These layers are then printed one by one, starting from the bottom layer and working up to the top layer (Figure 5c). The printer applies the material, usually in a liquid or powdered form, in the precise shape of each layer. The layers are then fused together, either through heat or chemical means, to create a solid object (Figure 5d) [126]. The process of 3D printing is a popular method for fabricating PEGDA hydrogels with controlled homogeneity in cross-linking and mechanical properties compared to bulk cross-linking (Figure 5e) [127,128]. Techniques such as extrusion-based printing [129], SLA, and DLP printing offer precise control over the hydrogel’s shape and properties and allow for the rapid creation of complex shapes [72,108,119,130]. SLA is a highly advanced technique for 3D printing hydrogels, using a layer-by-layer approach and UV to control photo-cross-linking [100,131]. The utilization of DLP in PEGDA hydrogel compositions, with different water content and concentrations of a photoinitiator and light absorber, demonstrated encouraging prospects for custom implant solutions tailored to individual patients [132]. Another study investigated DLP 3D printing of PEGDA-based photopolymers with dexamethasone (DEX) drug-loading, finding that drug release is prone to burst-release and DEX degrades significantly in the 3D-printed samples over time [133].

**Figure 5 polymers-15-02341-f005:**
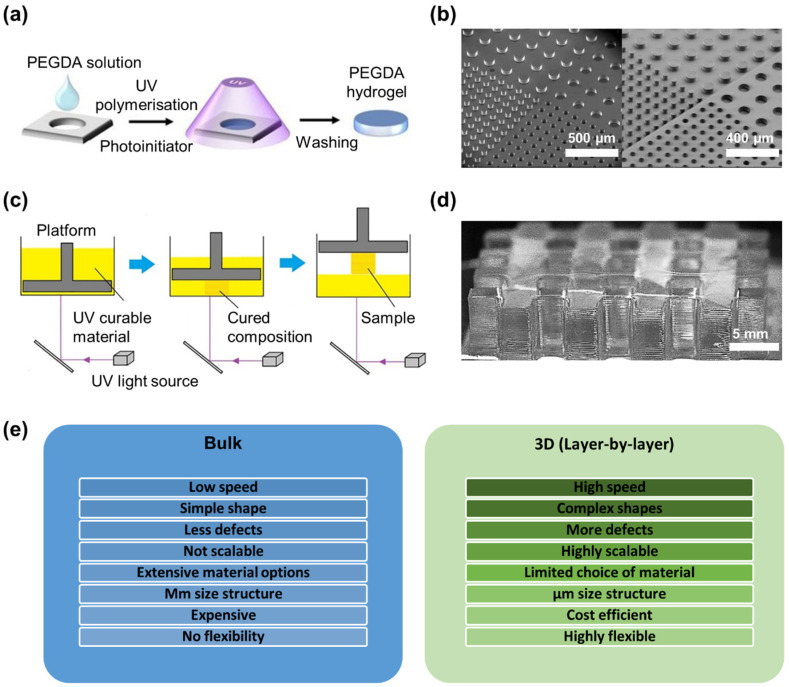
(**a**) Schematic illustration of synthesis of PEGDA hydrogels using a mould technique [134]. Reproduced with permission from ref. [134]. Copyright 2016 Elsevier. (**b**) SEM photographs illustrate a comparison between a Silicon master mould (**left**) and cross-linked PEGDA (**right**) in replicating low aspect ratio structures with 50 and 100 µm diameter and 50 µm height. [121]. Reproduced with permission from ref. [121]. Copyright 2012 Royal Society of Chemistry. (**c**) Schematic of the typical 3D printing techniques using a layer-by-layer printing [135]. Reproduced with permission from ref. [135]. Copyright 2020 Elsevier. (**d**) Optical microscopy image of grid shaped PEGDA hydrogel fabricated using 3D printing technique. (**e**) Advantages and disadvantages of bulk and 3D fabrication processes.

The ability to control the elasticity, swelling, and degradation of hydrogels is crucial in biomedical applications, particularly in tissue engineering. Photo-cross-linkable hydrogels offer the ability to mimic the dynamic nature of cellular microenvironments, providing a valuable tool for studying cellular mechanisms.

## 3. Properties of Photo-Cross-Linked PEGDA Hydrogels

### 3.1. Swelling Behavior and the Effect of the Environment

Hydrogels are a type of polymer that have the ability to absorb and retain large amounts of water, which leads to the formation of a gel-like structure (Figure 6a). Their extensive usage in biomedical applications stems from their biocompatibility and capacity to replicate the mechanical characteristics of natural tissue. (Figure 6b). The physical properties of hydrogels can vary depending on the type of polymer and the conditions under which they are prepared. The structural and physical properties of the hydrogels can be significantly influenced by adjusting the molecular weight and concentration of PEGDA in the prepolymer solution. (Table 1) [136].

Increasing the molecular weight of PEGDA from 3.4 kDa to 20 kDa in the prepolymer solution while maintaining a 10% precursor concentration, led to a 101% increase in swelling ratio and 190% increase in mesh size, resulting in higher cell proliferation [137]. To the contrary, an increase in PEGDA concentration led to a decrease in hydrogel swelling, mesh size, and limited vessel invasion, which may be required for neovascularization [138]. UV irradiation of PEGDA solution with different UV dosage resulted in hydrogels with a varying cross-link density. Increased UV exposure time led to a decrease in swelling capacity due to enhanced cross-linking and increased elasticity of the polymer chains [139]. The swelling ratio of the PEGDA hydrogel decreases as the ionic strength within the physiological range (0–300 mM) increases, while the hydrogel’s swelling ratio gradually increases across the entire pH range tested (2.5–11), with an approximate 10% increase [139]. PEG hydrogels are often used in short-term, in vitro studies as non-degradable controls due to the hydrolytic stability of the poly(ether) backbone. A 6-week study of storage of PEGDA hydrogels in PBS showed a slight increase in swelling ratio of 5–10% but it was within the standard deviation, indicating no degradation [140].

Fabrication techniques used for making the hydrogel can also affect the swelling kinetics of hydrogels. Studies indicate that 3D printed GelMA/PEGDA hydrogels created through extrusion-based printing demonstrate an approximately 10% higher swelling ratio than bulk cross-linked hydrogels. This enhancement can be attributed to the presence of pores within the 3D printed hydrogel matrix, which expands the surface area exposed to the PBS solution and consequently leads to increased water absorption [141]. In an extrusion based printed PNIPAM/PEGDA bilayer, the PEGDA composite fibre was reported to have a swelling ratio in the range of 0.1–3 [129]. In SLA printing, a variety of 3D printed PEGDA hydrogels with different molecular weights and concentrations have been synthesized in combination with other polymers such as PCL, PEGMEMA, and PEDOT for biomedical applications. The swelling ratios of these hydrogels range from 0.03 to 73, as reported in studies [72,142,143]. The values of swelling ratio reported in literature may vary due to the different methods used to calculate them (Table 1); some use percentage increase from dried hydrogel [144,145,146], others do not multiply the results by 100 [147,148,149] and others calculate by dividing the swollen state hydrogel by the dried hydrogel [129,150]. Although the swelling ratio of PEGDA hydrogels can vary across different types, a comparison between bulk and 3D printed PEGDA hydrogels at a fixed molecular weight of 0.7 Da reveals a consistent trend. However, it is noteworthy that the 3D printed hydrogels exhibit higher swelling ratios, indicating that the fabrication technique can significantly influence the swelling behavior of PEGDA hydrogels.

Examining the volumetric change as a result of swelling and water uptake is crucial when it comes to printing resolution and fidelity of the printed parts (Figure 6c). A study found that dog-bone shaped 3D printed PEGDA (575 Da) had a swelling ratio of approximately 35%, but there was a dimensional change of 15% in length/width and 40% in thickness [146]. The volumetric swelling ratio of bulk PEGDA hydrogel cylinders with molecular weights of 10 kDa and 20 kDa, determined through optical microscopy measurements of the hydrogel’s physical dimensions, was observed to be non-proportional to the gravimetric swelling ratio obtained by weighing the samples [151]. This means that while the swelling ratio indicates a change in volume, the volumetric percentage change may be different from the swelling, as the dimensional changes in length/width and thickness are also considered. A study that focused on microfluidic channel fabrication found that the mixture of water-miscible acrylate (HBA), PEGMEMA, and PEGDA resulted in internal cross-sections as small as 71 µm, which is smaller than the intended internal diameter of 108 µm. This was due to the swelling and volume change of the hydrogel material during the fabrication process [152]. To optimize the dimensional accuracy of 3D printed PEGDA hydrogels, a study varied UV exposure time and PEGDA concentration. Results showed that increasing UV exposure decreased accuracy and as PEGDA concentration increased from 2% to 80%, the percentage change in lateral dimensions while swelling also increased from 1% to 11% [153]. The volumetric degree of swelling can also be calculated indirectly by measuring the swelling ratio while taking into account the solvent and polymer density. However, that approach might not be as accurate because it does not directly measure the volume change but assumes constant solvent density and the density of the initial prepolymer solution which may change due to incomplete cross-linking after the initial photo-cross-linking process [151,154].

It is also important to understand the swelling kinetics of PEGDA hydrogels in cell culture media within the relevant physiological environment [155]. In many studies, the use of phosphate-buffered saline (PBS) at a pH level of 7.4 was implemented to replicate the necessary physiological conditions for cell culture [54,156,157,158,159]. It was reported that a combination of Poly(ε-caprolactone) maleic acid (PGCL-Ma) and PEGDA produced by bulk cross-linking has resulted in no degradation when stored for 20 days [144]. Introduction of PEGDA in a PEGDA/GelMA hydrogel also has been shown to reduce the enzymatic degradation profile of the hydrogel photo-cross-linked with UV light in PBS [141,148]. However, a bulk photo-cross-linked PEGDA hydrogel on its own has shown to undergo a very slow degradation when stored in BPS at 37 °C for a duration of 8 weeks [160]. The slow degradation of ester linkages in acrylated PEG diols in vivo can take months or even years [140]. Nevertheless, the degradation of PEGDA hydrogel can be considerably accelerated by higher molecular weights and lower concentrations, primarily due to elevated ratios of hydrolysable esters and reduced cross-link density [156].Here, we prioritize the in vitro degradation of the PEGDA hydrogels, as it is more relevant to the application of organ-on-chip devices in tissue engineering. The main focus is to understand the expected physical, chemical, and mechanical stability for such applications. However, future studies may require a separate investigation of in vivo applications.

Temperature can affect the swelling kinetics of PEGDA hydrogels. While a study reported insensitivity to changes in temperature and solvent composition, it was found that bulk cross-linked PEGDA hydrogels had slightly higher swelling ratios when incubated in deionized water at 37 °C compared to room temperature [129,139]. However, a recent study that looked at the reversible swelling/deswelling mechanism using environmental scanning electron microscopy (SEM) showed that depending on the degree of cross-linking, the 3D printed PEGDA fabricated by SLA can revert back to close to its original volume when the temperature varies between 4–15 °C [161]. This phenomenon occurs due to the hydrogel’s absorption of condensed water, resulting in an expansion of its volume at low temperatures. Conversely, as the temperature rises, the water is desorbed from the hydrogel, causing the volume to decrease. This behavior is typically observed when the samples are not fully immersed in the storage environment.

**Figure 6 polymers-15-02341-f006:**
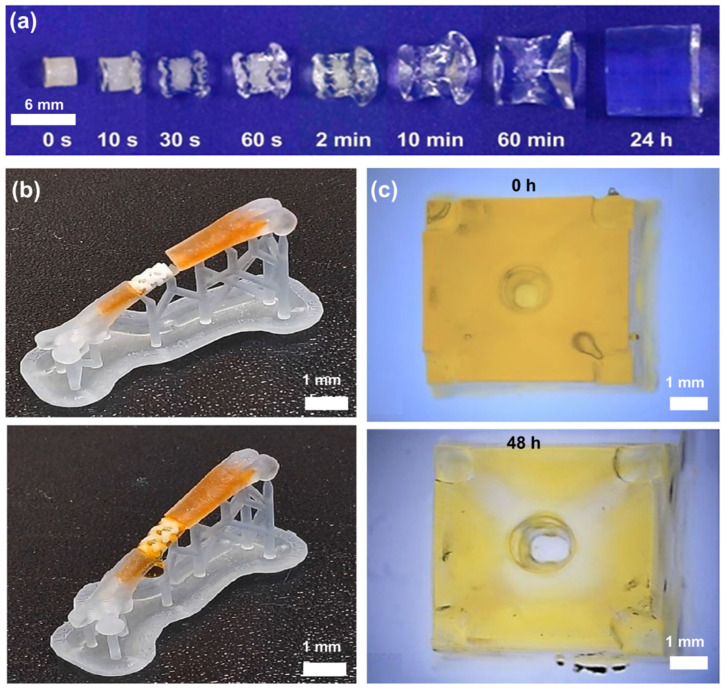
Swelling behavior of PEGDA hydrogels at different scale and shape complexity. (**a**) Side view of swelling of bulk polymerized PEGDA hydrogel in at 37 °C for first 24 h [162]. Reproduced with permission from ref. [162]. Copyright 2017 American Chemical Society. (**b**) Images of a complex brushite/PEGDA hydrogel biocomposite implant that can be implanted into a bone defect and fixed through swelling. The biocomposite is initially placed freely in the dry state (**top**), but swells and tightly fills the defect within 30 min by absorbing a red-colored aqueous solution containing an azo dye (**bottom**) [163]. Reproduced with permission from ref. [163]. Copyright 2020 Elsevier. (**c**) The 3D-printed samples stored in water for 48 h, showing an increase in inner channel and overall structural dimensions [164]. Reproduced with permission from ref. [164]. Copyright 2021 Springer Nature.

The properties of hydrogels depend on the type of polymer and the chemical and physical bonds between polymer chains. Ideally, cross-linked hydrogels have better mechanical properties than non-ideally cross-linked ones, but the effect of these cross-linking sites on water and polymer interaction is not well understood [165,166]. The photo-cross-linking process can be hindered by factors such as oxygen inhibition, which can lower the conversion rate of the prepolymer solution. Acrylates are particularly sensitive to oxygen inhibition, and the dissolved oxygen in the solution must be completely consumed before cross-linking can begin, leading to a slower reaction rate [161]. PEGDA (CH2=CHCO(OCH2CH2)nOCOCH=CH2) is a type of PEG that has repeated ethylene oxide units and active end groups which can be cross-linked by exposing it to a light source and a photoinitiator [167]. FTIR spectroscopy was used to study the surface chemical composition of bulk PEGDA membranes with varying PEGDA monomer concentrations. The spectra for all cross-linked samples were mostly identical, indicating a similar chemical composition [167]. The gel fraction was used as a complementary technique to qualitatively evaluate the efficiency of the network formation or degree of cross-linking [168]. In a separate investigation, the chemical structure was analyzed using proton nuclear magnetic resonance (^1^H NMR) spectroscopy, revealing that the C=C of PEGDA of PEGDA monomers were photo-cross-linked in the presence of a photoinitiator [169]. Measuring the peak areas of C=C and C-O-C bonds provides an indirect method to qualitatively assess the degree of cross-linking or conversion of the double bonds in PEGDA hydrogel, indicating the extent of acrylate group conversion [155]. Understanding the degree of cross-linking can aid in understanding how it affects the microstructure, porosity, and mechanical properties of PEGDA hydrogels. The following section discusses the different characterization tools used for understanding the mechanical properties of PEGDA hydrogels with more focus on SLA fabricated samples.

**Table 1 polymers-15-02341-t001:** Swelling characteristics of PEGDA with/without comonomer.

PEGDA/Comonomer	Molecular Weight (kDa)	Conc (%)	Bulk/3D	Type of Study	Cells Used	Sample Type	Storage Solution	pH	Temp (°C)	Volumetric Change Yes/No	Duration (Day)	Degradation	Swelling Ratio	Ref.
PEGDA	0.7	20	3D-projection lithography	Materials only	* N/a	Cuboid, 5 × 5 × 10 mm, 20 and 150 µm layer thicknesses	Deionized water	7.4	8–45	Yes	7	No	(−)10–(+)24	[56]
PEGDA	0.7	20	3D-projection lithography	Materials only	N/a	Cuboid, 5 × 5 × 5.1 mm, 20 and 150 µm layer thicknesses	Deionized water/cell culture media	7.4	8–37	Yes	30	No	(−)6–(+)20	[57]
PEGDA/poly-(ε-caprolactone) (PCL)	1	15	3D-SLA-DLP	In vitro viability of tissue constructs	Human umbilical vein endothelial cells (HUVECs)	n/a	Water	n/a	RT	No	3	No	5	[72]
PEGDA	0.7, 3.4, 5, 10	40	3D-SLA	Long-term viability of SMT	NIH/3T3 cells	Disks 100 µm, dia 5 mm, 1 mm thickness	PBS	7.4	37	No	1	No	5–35	[100]
Poly-N-isopropylacrylamide (PNIPAM)/PEGDA	N/a	10	3D-extrusion	Materials only	N/a	Multilayer, 260 µm	Water	N/a	RT	Yes, reversible between 20–50 °C	1	No	0.1–3	[129]
GelMA/PEGDA	0.7	0–15	Bulk and 3D-extrusion	Drug delivery	N/a	Multilayer, 200 µm,	PBS	7.4	36.1	No	1	Yes, 7 days	4–7	[141]
PEGDA/poly(ethylene glycol) methyl ethyl methacrylate (PEGMEMA)	0.25, 0.575	45–100	3D-SLA	In vitro Viability of cell encapsulation in micro-stereolithography	HUVECs	Multilayer 50 µm, Φ9 mm, 1 mm	PBS	N/a	RT	No	1	No	0.03–0.74	[142]
PEGDA/poly(3,4-ethylenedioxythiophene) (PEDOT)	0.575	50, 70, 90	3D-SLA	Materials only	N/a	Multilayer	PBS	N/a	37	No	1	No	0.13–0.35	[143]
Poly(ε-caprolactone) maleic acid (PGCL-Ma) and PEGDA	2, 3.6, and 8	0–100	Bulk	In vitro release of bovine serum albumin (BSA) for drug delivery	N/a	No specific shape, a section 25 mg	PBS	7.4	37	N/a	40	No degradation for 20 days	4.1–73	[144]
PEGDA/gelatine methacrylate (GelMA)	0.7	5-20	Bulk	Cell-laden cartilage tissue construct	Human bone marrow MSCs	Nanospheres	PBS	N/a	37	No	N/a	N/a	6–10	[145]
PEGDA/SiO_2_	0.575	N/a	3D-SLA	Materials only	N/a	Multilayer 25, 50, 100 µm (145 × 145 × 175 µm)	Water	N/a	25	Yes, 22% in thicknees and 15% on width	4	No	0.35	[146]
PEGDA	N/a	10	Bulk	Materials only	N/a	Cylinder, diam 3 mm, 15 mm thickness	Deionized water	N/a	37	No	3	N/a	5–13.5	[147]
PEGDA/GelMA	1, 4, 8	15, 20, 30	Bulk	3D cell culture platform for studying cell invasion	MDA-MB-231 cells	N/a	PBS	4, 7, 7.4	37	No	3	No degradation for 21 days	10–12	[148]
PEGDA/tendon tECM	N/a	10	3D-SLA	Tendon extracellular matrix for bone regeneration	Mesenchymal stem cell	Multilayer	Water	N/a	25	No	1/2	No	0.06–0.12	[150]
Water-miscible acrylate (HBA)/poly(ethylene glycol) methyl ethyl methacrylate (PEGMEMA)/PEGDA	0.575, 0.7,	2.5,7.5	3D-SLA	Materials only	N/a	Multilayer, 50 µm, dia 9 mm, 0.8 mm thickness	Mili-Q waer and PBS	N/a	RT	Yes	3–24 h	No	0.1–0.8	[152]
PEGDA	0.575	2–80	3D-DLP	Materials only	N/a	Multilayer, 50, 100, 500 µm,	Deionized water	N/a	RT	Yes	1	No	0.01–0.2	[153]
PEGDA	0.7, 3.4	10,20,40	Bulk	In vitro 3D synthetic matrices	NIH 3T3 cells	Cylinder, dia 10 mm, thickness 1 mm	Deionized water	N/a	RT	No	N/a	No	2–9	[155]
GelMA-PEGDA	20	10	Bulk	In vitro 3D hydrogels to mimic the neural tissue	Mouse neuroblastoma cell line Neuro2a	Cylinder, dia 10 mm, 3 mm thickness	Collagenase type II	N/a	RT	No	2	Yes,	0.01–0.8	[156]
PEGDA	0.575	N/a	3D-SLA	Materials only	N/a	Multilayer, 100 µm, 200 × 200 µm	Water	N/a	4–15	Yes, reversible depending on cross-linking	N/a	N/a	0.2–2.3	[161]
PEGDA	10	7.5	Bulk	Materials only	N/a	Tubes with diameters 3.85–7.2	Mili-Q	7.4	37	Yes	2	No	30	[162]
Calcium phosphates (Cap)/PEGDA	0.250, 0.575, 0.7	N/a	3D-SLA	In Vivo 3D bone grafts	N/a	Multilayer. 6 mm dia, 200 µm thickness	Water	N/a	RT	No	7	No	0.20–0.60	[163]
PEGDA	0.575	N/a	3D-SLA	Materials only	N/a	Multilayer, cantilevers	Ethanol, Water, acetone	N/a	RT	N/a	60 min	no	1.2–1.7	[170]
PEGDA/Acrylic acid (AA)	1, 4, 10	9–36	Bulk	In vitro contractile SMT	C2C12 mouse myoblast cell	Single layer, 0.4 mm thickness	PBS	7.4	RT	Yes	3	Yes, after 4 weeks	1–3.5	[171]
PEGDA	0.7	20	3D-SLA	In vitro contractile SMT	C2C12 mouse myoblast	Multilayer, 20 um, dia 6 mm, 5 mm thickness	Deionized water	N/a	RT	No	1	No	5.1–5.6	[149,154]

* N/a: No information is available or provided in the study.

### 3.2. Mechanical Properties of PEGDA Hydrogels

Methods have been developed to assess the elastic modulus of in vitro scaffolds, particularly hydrogels, which are widely utilized in 3D culture due to their tunable characteristics and compatibility with biological systems, with the goal of achieving a 3D matrix that accurately replicates the mechanical properties of the tissue or pathological environment under study [172,173].

#### 3.2.1. Bulk Characteristic of PEGDA Hydrogels

Soft biological tissues typically demonstrate an increasing resistance to deformation with the rise in applied stress, which is commonly evaluated using the tensile test to measure the Young’s modulus as the ratio of applied stress to resultant strain (Figure 7a). When conducting tensile tests on biological samples, it is crucial to measure the strain at a specific stress while assuming the material behaves as a perfectly elastic one (Figure 7b) [174]. The 3D printed PEGDA hydrogel using SLA tested on tensile machine reported an elastic modulus of 1.4 MPa [175]. Combining hydrogels with other polymeric materials can be a viable approach to overcome the inherent limitation of low mechanical stability exhibited by hydrogels under physiological conditions [176]. In the same study, blending the PEGDA with tetraethoxysilane nanoparticles increased the modulus by at least 50% [175]. Another study showed that a bilayer of PNIPAN/PEGDA produced by extrusion-based 3D printing had a Young’s moduli of 36 kPa [129]. Even though the mechanical properties of PNIPAN/PEGDA were considerably low, the addition of PNIPAN has resulted in having a thermoresponsive composite hydrogel that reacts differently at temperatures ranging from 20 to 50 °C. A study that used a combination of PEGDA/GelMA hydrogels produced through bulk cross-linking for researching neurodegeneration found that the modulus measured during a tensile test was in the range of 10 kPa. This value is consistent with the range required for neural stem cell therapies, which is typically between 1–7 kPa [156,177].

The compression test is another frequently employed method to evaluate the elastic modulus of PEGDA hydrogels, requiring well-defined cylindrical samples with specific shapes and dimensions (Figure 7a,c) to ensure precise and reliable results, similar to tensile tests [178]. Multiple studies found that PEGDA hydrogels created by both bulk cross-linking and 3D printing with similar molecular weight of 700 Da had compression moduli in the range of 0.02 to 1.29 MPa. The compressive modulus has an inverse relationship with polymer molecular weight and a direct relationship with polymer concentration (Figure 7d,e) [100]. A bulk characterization of GelMA/PEGDA (700 Da) hydrogel using a compression test showed a compressive modulus of 5–20 MPa. A combination of GelMa/PEGDA (20 kDa) also produced by bulk cross-linking reported a considerably lower compressive modulus compared to GelMA/PEGDA (700 Da), as low as 0.8 KPa [156]. Such a change in compression modulus could also be due to the different strain rate used in the studies [179]. A composite of 3D printed PEGDA hydrogels with SiO_2_ nanoparticles using SLA had a compressive modulus of 50–85 MPa which is considerably higher than what was reported in literature review [146]. Such nanocomposites with enhanced mechanical properties can be explained by the high surface area-to-volume ratio surface interaction between nanofillers and polymers [180].

**Figure 7 polymers-15-02341-f007:**
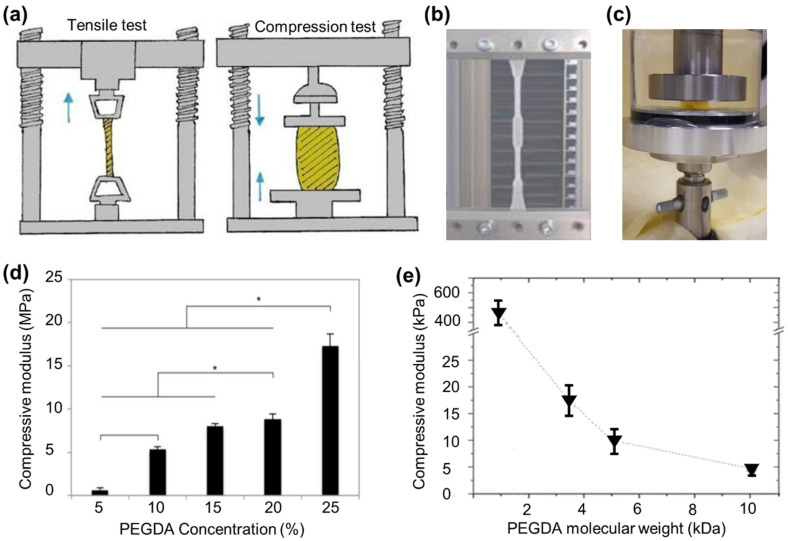
(**a**) Schematic of tensile and compression. Blue arrows illustrate the direction of force [181]. Reproduced with permission from ref. [181]. Copyright 2017 Elsevier. (**b**) Image of PEGDA hydrogel under tensile tension [116]. Reproduced with permission from ref. [116]. Copyright 2022 De Gruyter. (**c**) Image of 3D PEGDA hydrogel under compression. (**d**) Compressive moduli of hydrogels with varying PEGDA concentrations. n = 5, * *p* < 0.05. Error bars represent standard deviation [145]. Reproduced with permission from ref. [145]. Copyright 2018 IOP Publishing. (**e**) Mechanical properties of laser-polymerized PEGDA hydrogels were measured for 20% PEGDA hydrogels as a function of molecular weight (0.7, 3.4, 5, and 10 kDa) [100]. Reproduced with permission from ref. [100]. Copyright 2010 Royal Society of Chemistry.

The mechanical properties of PEGDA hydrogels produced using photo-cross-linking can be altered by changing the UV light dosage. Increasing the UV light dosage may result in a decrease in the compressive modulus due to over-cross-linking. This was observed in studies where PEGDA hydrogels were exposed to different UV light durations. This was observed in studies where PEGDA hydrogels were exposed to 20, 40, and 60 s of UV light. The compression modulus increased from 0.49 to 0.7 MPa when exposed to 20 and 40 s, respectively, but dropped to 0.64 MPa when exposed to 60 s of UV light [147].

The findings presented in Table 2 indicate that both bulk and 3D printed PEGDA hydrogels exhibit strain-dependent behavior, suggesting that they are viscoelastic materials. However, further investigation is needed to fully understand the mechanical properties of these hydrogels. It is worth noting that conventional compression or tensile testing methods, typically used for bulk characterization, may not provide an accurate representation of the PEGDA system. Therefore, in order to gain a better understanding of these 3D printed systems, the utilization of more specialized techniques such as rheology and dynamic mechanical analyzer (DMA) is warranted. Although measurements using these techniques have been conducted on both bulk and 3D PEGDA hydrogels, it is important to note that these techniques are beyond the scope of the present research. Nonetheless, future investigations should consider employing rheology and DMA to further elucidate the mechanical properties of 3D printed PEGDA hydrogels. Moreover, the variation in reported elastic properties emphasizes the necessity of establishing a standardized protocol specifically tailored for hydrogels. Given the established application of PEGDA hydrogels in the biomedical field, particularly in their interaction with living tissues and cells, it is essential to consider that the mechanical properties of 3D printed cell-laden cartilage constructs, which have been assessed using bulk samples, might not accurately reflect the mechanical behavior at a smaller scale. This discrepancy arises despite the material’s intended use in 3D bioprinting. Therefore, it becomes imperative to explore alternative testing approaches that can provide more accurate representations of the mechanical properties at the desired scale, ensuring that the performance of PEGDA hydrogels aligns appropriately with their biomedical applications [145,163]. To obtain accurate surface mechanical properties at the micro and nanoscale, advanced techniques such as AFM and nanoindentation are commonly employed in various studies [182]. Although indentation testing may produce comparable results to compression testing in terms of the elastic modulus, it differs significantly due to differences in loading geometry and boundary conditions [183,184,185].

**Table 2 polymers-15-02341-t002:** Bulk characterization of PEGDA hydrogels.

PEGDA/Comonomer	Molecular Weight (kDa)	Conc (%)	Bulk/3D	Type of Study	Cells Used	Sample Shape	UV Variations	Storage Solution	Temperature (°C)	Mechanical Test	Modulus	Ref.
PEGDA/poly-(ε-caprolactone) (PCL)	1	15	3D-SLA-DLP	In vitro viability of tissue constructs	HUVECs	Disks, 100 µm, dia 8 mm, 3 mm thickness	Yes	* N/a	N/a	Compressive modulus, 1 N load cell, 0.1–30% strain	50–250 kPa	[72]
PEGDA	0.7, 3.4, 5, 10	40	3D-SLA	Long-term viability of SMT	NIH/3T3 cells	Disks 100 um, dia 5 mm, 1 mm thickness	No	PBS	25	Compressive modulus, compression test, 1 mm/s, strain at 10%	5–503 kPa	[100]
Poly-N-isopropylacrylamide (PNIPAM)/PEGDA	N/a	10	3D-extrusion	Materials only	N/a	Multilayer, 260 µm,	No	Solution	25	Tensile strength, uniaxial tension, loading rate 2 mm/s	85 kPa	[129]
PEGDA/gelatine methacrylate (GelMA)	0.7	5–20	Bulk	Cell-laden cartilage tissue construct	Human bone marrow MSCs	N/a	No	Dry	RT	Compression/100 N load, 2 mm/min	5–20 MPa	[145]
PEGDA/SiO_2_	0.575	N/a	3D-SLA	Materials only	N/a	Multilayer 25, 50, 100 µm (145 × 145 × 175 µm)	No	Swollen (water)	25	Compression testing, 2 mm/min	50–85 MPa	[146]
PEGDA	N/a	10	Bulk	Materials only	N/a	Cylinder, diameter 15 × 3 mm thickness	Yes	Dry	RT	Compression/ 2 kN, 0.5 mm/min	0.49–0.7 MPa	[147]
PEGDA/GelMA	1, 4, 8	15, 20, 30	Bulk	3D cell culture platform for studying cell invasion	MDA-MB-231 cells	N/a	No	Hydrated (water)	37	Storage modulus, rheometry, 1 to 400 rad/s, amplitude of 0.3%.	1–8 kPa	[148]
PEGDA/tendon tECM	N/a	10	3D-SLA	Tendon extracellular matrix for bone regeneration	Mesenchymal stem cell	Multilayer	No	Dry	RT	Compression, compressive modulus	0.2–0.3 MPa	[150]
PEGDA	0.7, 3.4	10, 20, 40	Bulk	In vitro 3D synthetic matrices	NIH 3T3 cells	Cylinder, dia 10 mm, thickness 3 mm	No	Swollen (water)	RT	Confined compression, 19 µm/min, 10% deformation	22–118 kPa	[155]
GelMA-PEGDA	20	10	Bulk	In vitro 3D hydrogels to mimic the neural tissue	Mouse neuroblastoma cell line Neuro2a	Tensile 30 × 10 × 3 mm, compression dia 10 mm, 5 mm thickness	No	PBS	37	Tensile modulus, max strain at 50%, 1mm/min compressive modulus, 0.5 mm/min	10–60 kPa 0.8–6 kPa	[156]
Calcium phosphates (Cap)/PEGDA	0.250, 0.575, 0.7	N/a	3D-SLA	In vivo 3D bone grafts	N/a	Multilayer. 200 um. 6 mm dia, 12 mm thickness	No	Dry and swollen (water)	25 and 37	Compression testing, 1 mm/min	0.21–19.87 MPa	[163]
PEGDA/Acrylic acid (AA)	1, 4, 10	9–36	Bulk	In vitro contractile SMT	C2C12 mouse myoblast cell	Single layer, 2 × 5 cm 0.4 mm	No	PBS	RT	Tensile test, elastic modulus, 1.5 mm/min	68–214 kPa	[171]
PEGDA	0.6	N/a	3D-DLP	Materials only	N/a	10 × 80 × 0.7 mm	No	N/a	N/a	Tensile test, Young’s modulus, load cell 500 N	1.4 MPa	[175]
PEGDA	0.258	7–40	Bulk	Human articular cartilage	N/a	Cylinder, dia 11 mm, height 8 mm	No	Dry wet	RT	Tensile test, elastic modulus, 20 N, compression test, elastic modulus	4–20 MPa 0.05–3.19 MPa	[182]
PEGDA	0.7	20	3D-SLA	In vitro contractile SMT	C2C12 mouse myoblast	Multilayer, 20 um, dia 6 mm, 5 mm thickness	No	Swollen (water)	RT	Compression testing shear modulus, 0.5 mm/min	0.43 MPa	[149,154]

* N/a: No information is available or provided in the study.

#### 3.2.2. Localized Characteristics of PEGDA Hydrogels

Nanoindentation and AFM are extensively utilized techniques for assessing the mechanical properties of soft biological tissue materials and optical thin films. They are particularly beneficial for testing hydrogels, as they require only small samples, can be performed on a single sample, and reduce differences between samples. Both techniques involve pressing a probe onto the surface of the hydrogel and creating a force-indentation distance curve, from which the hydrogel’s Young’s modulus can be obtained [186,187]. The probe characteristics, such as material, spring constant, and radius, have a significant impact on the measurement’s resolution, accuracy, and sensitivity [188,189,190]. AFM has been utilized to examine the elastic modulus of PEG based hydrogels (Figure 8a) [191,192,193]. For instance, AFM results for PEGDA hydrogels with varying molecular weight showed an increase in elastic modulus [194], contradicting bulk compression test results of an inverse relationship between the molecular weight and compression modulus [155]. Therefore, it could possibly be a difference in results obtained in macroscale such as compression testing and nanoscale such as AFM. The elastic modulus of bulk-cross-linked PEGDA hydrogels increased from 30 to 80 kPa as the concentration of the PEGDA monomer increased from 20 to 50%, as determined by AFM analysis (Figure 8b) [195]. The mechanical properties of 3D printed PEGDA hydrogels produced by SLA and direct laser writing (DLW) were analyzed using AFM. The results showed that the elastic modulus of these hydrogels was in the range of 1-10 MPa (Figure 8c) [161,196]. A longer UV light dosage results in higher elastic modulus as it dictates the degree of cross-linking. AFM measurements on 3D PEGDA hydrogel show that elastic modulus increases linearly with increased UV light dosage and increased degree of cross-linking (Figure 8d) [195].

**Figure 8 polymers-15-02341-f008:**
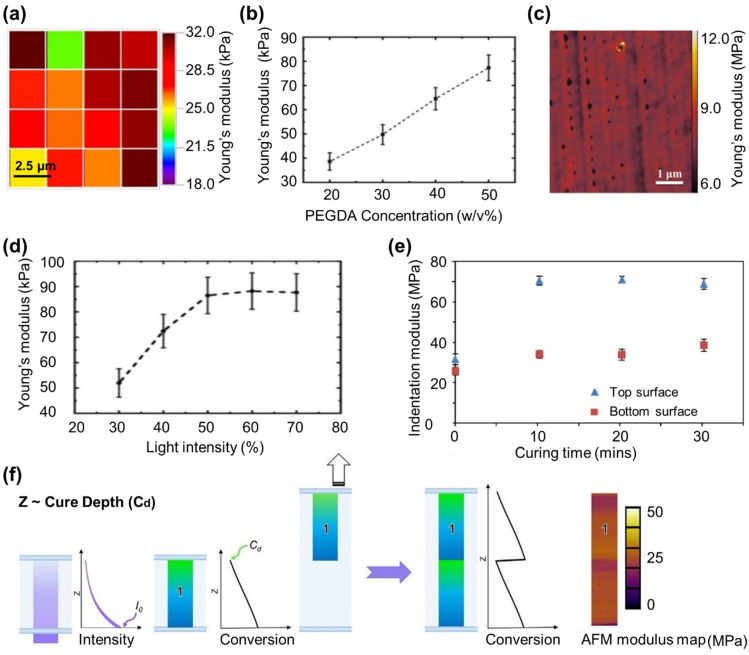
Localized mechanical properties of PEGDA hydrogels measured using AFM and nanoindentation techniques. (**a**) Elastic modulus map of bulk hydrogels showing variations between 18–32 kPa [193]. Reproduced with permission from ref. [193]. Copyright 2011 Elsevier. (**b**) Young’s modulus of a hydrogel with change in PEGDA concentration between 20 to 50 *w*/*v*% measured using AFM [195]. Reproduced with permission from ref. [195]. Copyright 2022 Elsevier. (**c**) AFM topographic image of 3D hydrogels produced using DLW using PeakForce QNM mode showing elastic modulus ranging between 6 to 12 MPa in dry state [196]. Reproduced with permission from ref. [196]. Copyright 2021 Elsevier. (**d**) The influence of light intensity on the mechanical properties of hydrogel is examined by AFM, showing an overall positive trend [195]. Reproduced with permission from ref. [195]. Copyright 2022 Elsevier. (**e**) Plots of nanoindentation for 3D printed hydrogel using inkjet printing showing the difference in nanomechanical properties at top and bottom of the hydrogel after postcuring process. The results show that unidirectional postcuring results in heterogeneity in elastic modulus across the sample [86]. Reproduced with permission from ref. [86]. Copyright 2016 Wiley. (**f**) Multilayer hydrogels produced by DLP printer with irradiance intensity (I0), and layer thickness (Z) parameters used to control cure depth (Cd). AFM used for mapping elastic modulus cross-sectioned of hydrogels showing how incorporating a light absorber, can result in exponential decay in light intensity with penetration depth, resulting in a limited thickness of resin that undergoes gelation [197]. Reproduced with permission from ref. [197]. Copyright 2021 Wiley.

Nanoindentation, as an alternative method, allows for the mapping of elastic properties of soft materials at a microscopic level by employing an indenter tip to apply a controlled force and precisely measuring the indentation depth with sub-nanometer accuracy [198]. This technique proves valuable in studying the influence of the micro-environment on the mechanical behavior of the matrix, with specialized flat punches and large-radius spherical indenters developed specifically for soft samples [95,199]. The elastic modulus of 3D PEDGA hydrogels used as a scaffold for bone marrow mesenchymal stem cells, produced by two-photon polymerization, increased from 100–1500 kPa as laser intensity and raster speed were increased [200]. In another study, 3D PEGDA structures were created using a technique called mask projected excimer laser SLA, without adding a photoinitiator to the prepolymer solution. This method relies on high levels of light energy and resulted in a significant increase in elastic modulus, as measured by nanoindentation [201]. The 3D PEGDA hydrogels produced by this technique without photoinitiator showed a 10-fold increase in elastic modulus when analyzed by nanoindentation [202]. Such advancement in 3D printing helps with reducing the risk of cytotoxicity from the photoinitiator molecules during cell proliferation [202]. Nanoindentation measurements provide insight into the time dependent behavior of hydrogels and how surface characteristics affect cell behavior when in contact with the hydrogel surface. These measurements have shown that hydrogels used for multicellular spheroid culturing exhibit time-dependent stress curves, indicating viscoelastic behavior that leads to better cell adhesion, spreading, and tissue formation [203]. Nanoindentation was used to compare the mechanical characteristics of the top and bottom surface of a 3D printed PCL/PEGDA hydrogel, revealing that the top surface had a higher elastic modulus than the bottom surface (Figure 8e). This difference can be attributed to the reduced intensity of UV light reaching the bottom surface during 3D printing, resulting in a lower degree of cross-linking [86]. This suggests that 3D printing can result in structures with variations in cross-linking and mechanical properties. It was found that 3D printed PEGDA hydrogels produced by DLP printing have a mechanical gradient across the layers, as confirmed by measuring the elastic modulus along the surface of cross-sectioned 3D-printed structures (Figure 8f) [197]. Moreover, examining samples in their hydrated and dried state also showed to report different elastic modulus with dry samples always have a higher modulus in comparison with swollen hydrated samples [146,163]. Nanoindentation research on PEGDA hydrogels showed that the elastic modulus of a dried PEGDA hydrogel was 5 MPa, while the modulus of hydrated samples was slightly above 1 MPa [204]. In Table 3, the elastic modulus variations are presented from multiple studies, taking into account different printing parameters, fabrication methods, and starting prepolymer solutions. The analysis reveals that although all the studies focused on tissue engineering applications, a significant portion (58%) of them were conducted under dry conditions, which do not reflect the actual conditions in which the hydrogel samples will be used. This leads to an overestimation of the elastic properties of PEGDA hydrogels in these studies. Once again, this raises the discussion surrounding the necessity of developing a standard protocol specifically designed for measuring the mechanical properties of hydrogels. Such a protocol would help minimize the discrepancies in measurement results found in the literature, ensuring greater consistency and reliability for researchers worldwide.

**Table 3 polymers-15-02341-t003:** Localized characterization of PEGDA hydrogels using AFM and nanoindentation.

PEGDA/Comonomer	Molecular Weight (kDa)	Conc (%)	Bulk/3D	Type of Study	Cells Used	Sample Shape	UV Variations	Storage Solution	Temperature (°C)	Mechanical Test	Modulus	Ref.
PEGDA	0.7	20	3D-projection lithography	Materials only	* N/a	Cuboid, 5 × 5 × 1.2 mm, 20 and 150 µm layer thicknesses	Yes	Deionized water/cell culture media	RT	Nanoindentation	0.67–1.69 MPa	[57]
PEGDA	0.7	20	3D-projection lithography	Materials only	N/a	Cuboid, 5 × 5 × 3 mm, 20 µm layer thickness	Yes	Deionized water	RT	AFM	2.8–13.1 kPa	[61]
Polycaprolactone di-methacrylated (PCLDMA)/PEGDA	0.250	0–50	3D-Inkjet printing	Biocompatibility test	NIH3T3 fibroblasts	Multilayer, 5 µm, 100 layers, cuboid, 5 × 5 × 0.5 mm	Yes	N/a	RT	Nanoindentation, spherical indenter 50 µm radius	25–75 MPa	[86]
PEGDA	0.575	N/a	3D-SLA	Materials only	N/a	Multilayer, 100 µm, 200 × 200 µm	Yes	N/a	N/a	AFM, elastic modulus	1.25–4 MPa	[161]
PEGDA	0.6	N/a	3D-DLP	Materials only	N/a	Multilayer, 10–100 µm thickness	No	Dry	N/a	Nanoindentation, radius 1 µm diameter,	7 MPa	[175]
PEGDA	0.258	7–40	Bulk	Human articular cartilage	N/a	Cylinder, dia 11 mm, height 8 mm	No	Wet	RT	Nanoindentatio, flat punch. 54 µm diameter	1.84–3.29 MPa	[182]
PEGDA/Hyaluronic acid (HA)	0.575	N/a	Bulk	Influence of the network architecture	Murine L929 fibroblasts	N/a	No	No	25	AFM,	2.5–25 kPa	[193]
PEGDA	0.258, 0.575, 0.7	N/a	Bulk	Materials only	N/a	Single layer, 1.5 mm thickness	No	Water	N/a	AFM	2.48–4.33 MPa	[194]
PEGDA	0.7	20–50	Bulk	Materials only	N/a	Cuboid, 6 × 6 × 0.4 mm	Yes	N/a	N/a	AFM	35–95 kPa	[195]
PEGDA	0.7	N/a	3D-Direct Laser writing (DLW)	Materials only	N/a	Single layer,	Yes	No	N/a	AFM, elastic modulus	6.9–9.5 MPa	[196]
PEGDA	0.7	N/a	3D-DLP	Materials only	N/a	Multilayer, 10, 30, 100 µm	Yes	Dry	RT	AFM	9.5–34.5 MPa	[197]
PEGDA	0.7	50	3D-DLW	In vitro woodpile structures, model for leukemic disease	Bone marrow mesenchymal stem cells (BM-MSCs)	Multilayer, 100 × 100 × 50 µm	Yes	Water	RT	Nanoindentation, spherical 28 µm radius	100–1500 kPa	[200]
PEGDA (no photoinitiator)	0.7	N/a	3D-laser SLA	In vitro cell viability and biocompatibility	Chinese hamster ovarian cells (CHO)	Multilayer, 100 µm thickness	Yes	Dry	N/a	Nanoindentation, Berkovich tip	10–100 MPa	[202]
PEGDA	6, 20	15	3D-SLA	In vitro 3D microenvironment and viability test	Cardiac progenitor cells (hCPCs)	Multilayer, 250 µm layer thickness, 10 layers	No	Dry and wet	Water	Nanoindentation	5 MPa 1 MPa	[204]
PEGDA	0.575	5–20	Bulk	Materials only	N/a	Single layer, 70 µm	No	Water	20	AFM, 2.5 µm radius spherical tip	2.8–228.9 kPa	[205]
PEGDA	0.7	N/a	3D-SLA	Materials only	N/a	Multilayer, 600 um height, diameter 300 µm	No	N/a	N/a	AFM	200 MPa	[206]
PEGDA (PEG-fibrinogen)	0.4, 4	5–25	Microfluidic	Material stiffness and cell viability	human foreskin fibroblasts (HFFs)	Single layer,	No	N/a	N/a	AFM	0.7–50 kPa	[207]
Polyacrylamide (PAM)/PEGDA	0.7	5–16	Microfluidic	Materials only	N/a	Hydrogel particles	No	Water, dry	RT	AFM	2–7 kPa (hydrated) 25–180 kPa (dry)	[208]

* N/a: No information is available or provided in the study.

## 4. 3D PEGDA Hydrogel for Cardiac Tissue Engineering

As discussed previously, PEGDA hydrogel has been used for various applications within the field of tissue engineering [37,209,210]. In the area of organ-on-chip devices, it has been tested for 3D cell culture platforms such as skeletal muscle microtissues (SMT) [211,212,213]. Designing functional in vitro SMT involves two crucial elements: scaffold architecture and functionality. The scaffold should facilitate muscle differentiation and myogenesis leading to aligned, dense, and well-oriented myofibers. The scaffold’s functionality can be improved through artificial electrical stimulation promoting myogenic cell transformation into functional myofibers [214,215,216,217].

The popular scaffold design for contractile cardiac tissue resembles PDMS molded cantilever beams (posts), with tissue hanging from both ends. It evolved from 2D to representative 3D tissue studies and multiple commercial systems exist (Figure 9a–h) [218,219,220,221,222,223,224,225]. The technique measures the deflection of vertical posts caused by the contraction of the SMT. As the tissue construct is connected to the top of the posts, it exerts a force that brings the posts closer together, resulting in horizontal displacement. This displacement can be utilized to calculate the contraction force [226,227,228].

**Figure 9 polymers-15-02341-f009:**
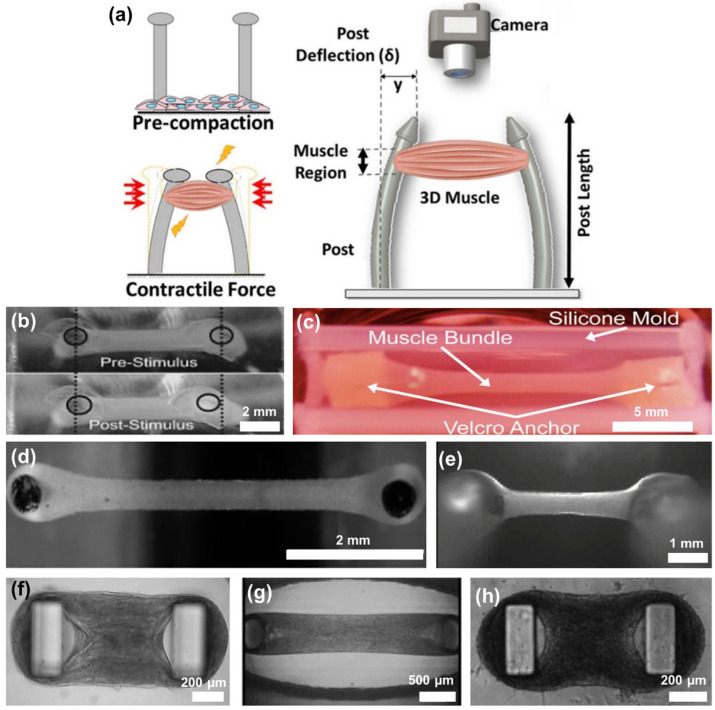
State-of-the-art MTS formation based on the quantification of the deflection of a pair of vertical cantilever PDMS beams (posts) for functional read-out assays. (**a**) Schematic of contractile force measurement for STM that is grown in vitro between flexible cantilevers (posts) serving as tendons. Force is quantified by tracking post displacements in response to stimulation and knowing platform mechanics [213]. Reproduced with permission from ref. [213]. Copyright 2022 eLife. (**b**) Molding process was employed to make caps for the posts, using primary mouse myoblasts, tissue strips were created encased in either a collagen/matrigel or fibrin scaffold [218]. Reproduced with permission from ref. [218]. Copyright 2008 Wiley. (**c**) Skeletal muscle bundle of rats myoblasts embedded with matrigel/fibrinogen and collagen I [219]. Reproduced with permission from ref. [219]. Copyright 2011 Elsevier. (**d**) Human engineered cardiac tissues (hECT) created using differentiated human pluripotent stem cells in multi-tissue bioreactor [220]. Reproduced with permission from ref. [220]. Copyright 2016 JoVE. (**e**) Atrial-like engineered heart tissue generated from human induced pluripotent stem cells (hiPSC) [229]. Reproduced with permission from ref. [229]. Copyright 2016 ICCSR. (**f**) Muscle tissue was created by embedding either C2C12 cells or iPSC-derived cardiomyocytes in a fibrin-based scaffold [230]. Reproduced with permission from ref. [230]. Copyright 2018 Elsevier. (**g**) The transformation of the ECM by human myoblasts forming a human muscle micro-tissue held by two posts for a 96-well culture platform [223]. Reproduced with permission from ref. [223]. Copyright 2020 Nature. (**h**) Miniaturized hiPSC-based cardiac tissue formed around micropillars [224]. Reproduced with permission from ref. [224]. Copyright 2020 IEEE.

As a better alternative to PDMS, which absorbs lipophilic compounds, PEGDA hydrogels limit nonspecific adsorption of proteins, DNA, and lipophilic drugs, making them suitable for drug testing devices [149]. Additionally, techniques such as masked photolithography and mold-based fabrication are both inefficient and unsuitable for large-scale production. These methods lack flexibility when it comes to creating complex shapes with reliable and repeatable mechanical properties that match living cells. On the other hand, 3D printing using micro-SLA and DLP offers a faster and more dependable manufacturing approach. These techniques involve the layer-by-layer fabrication of 3D PEGDA hydrogel posts. The concept originated from miniaturized walking robots that were constructed from a 3D-printed PEGDA hydrogel base and cantilever with two different molecular weights. Subsequently, the cantilever was seeded with contractile cardiomyocyte cells [231]. The cantilever was initially polymerized, then its base was 3D printed, and its surface was functionalized with collagen. The thickness of the cantilever was altered to assess its performance. Further development led to the creation of cantilevers with varying molecular weights, as a correlation between substrate stiffness and its impact on cardiomyocyte contractile behavior was established (Figure 10a,b) [232,233,234]. It was discovered that increasing the molecular weight of the PEGDA monomer in the prepolymer solution creates more compliant structures with elasticity similar to native myocardium, measured in kiloPascals [232]. Such knowledge led to the creation of a 3D printed flexible bio-bot that can hold a muscle strip together and respond to optical and electrical pulses [235]. The same printing technique was further developed to help investigate fabrication and the cryopreservation of skeletal MTS and the multicellular spinal cord-muscle bioactuator using 3D PEGDA hydrogels (Figure 10c) [131,236]. In another study the focus was to optimize 3D PEGDA hydrogel as an artificial muscle, focus was on adjusting its stiffness by changing the molecular weight and measuring its contractile strength and durability [171]. It was discovered that the hydrogels had a decline in performance after 4 weeks due to the gradual degradation of the material. In a most recent work, a 3D PEGDA microenvironment was printed to facilitate and support the formation of MTS [149]. This study investigated the impact of altering the diameter of 3D PEGDA micro-cantilevers and the concentration of PEGDA in the prepolymer solution on the behavior of MTS. It was observed that both increasing the diameter of the micro-cantilevers and raising the PEGDA concentration led to a decrease in the tissue’s ability to generate sufficient contractile forces for bending the cantilever. The force exerted by the tissue on the cantilever was calculated using the Euler-Bernoulli theory based on cantilever stiffness and deflection (Figure 10d) [149].

**Figure 10 polymers-15-02341-f010:**
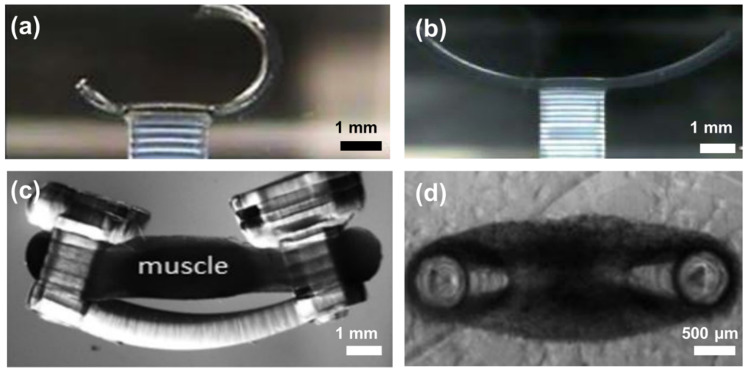
Images of 3D printed PEGDA cantilevers using SLA for contractile tissue studies. (**a**) Bio-bot 3D printed PEGDA cantilever with cardiac cell sheet seeded on bottom of the cantilever [231]. Reproduced with permission from ref. [231]. Copyright 2012 Nature. (**b**) Multi-material 3D PEGDA cantilever and base with cardiomyocytes seeded on the cantilever. Deflection occurs due to intrinsic stress in the formed tissue [232]. Reproduced with permission from ref. [232]. Copyright 2012 Nature. (**c**) Spinobot from the spinal cord and C2C12 myoblasts with tissue attached to the hydrogel skeleton which causes bending by generating passive tension [131]. Reproduced with permission from ref. [131]. Copyright 2020 AIP Publishing. (**d**) Muscle tissue created by embedding contractile cells of C2C12 hanging on 3D printed PEGDA hydrogels for multi assay platform [149]. Reproduced with permission from ref. [149]. Copyright 2019 American Chemical Society.

To achieve higher maturity in the MTS for enhanced cardiomyocyte research, a study proposed the integration of a flexible support structure made of the same material as the 3D printed PEGDA hydrogel beneath the cantilevers to enhance mechanical stimulation of the tissue for its maturity [237]. This was achieved by stretching the flexible structure, which in turn horizontally stretches the cantilever and mechanically contracts the MTS, thus promoting tissue maturation. In addition to mechanical stimulation, electrical and optical stimulation are widely recognized as key techniques for promoting maturity in mechanically-tensioned tissues [238,239,240].

## 5. Knowledge Gap and Future Perspective

Despite the development of advanced techniques such as SLA, DLP, microfluidic, and inkjet printing, there are still many gaps in our knowledge of 3D printed PEGDA hydrogels. One critical area that needs to be addressed is the impact of layer-by-layer fabrication on their behavior and properties, including their thermal response, isotropic behavior, cross-linking homogeneity, dimensional and mechanical stability, and the effect of printing parameters on their surface nanomechanical properties and viscoelastic behavior (Figure 11). All of these factors are crucial for the majority of organ-on-chip devices, especially in contractile force measurements, where the 3D PEDGA hydrogel pillars are used to apply or resist force [149].

For example, to accurately analyze the pillar bending in contractile force measurements, we must fully understand the hydrogel’s dimensional stability, physical isotropy, mechanical behavior, and viscoelasticity, which are crucial in determining the model used to predict the tissue force. Furthermore, the hydrogel’s dimensional stability resulting from changes in temperature should be thoroughly investigated to understand their thermal response fully. The effects of layer-by-layer printing on physical anisotropy, which can depend on the printing direction and direction of light, also need further exploration.

Moreover, the impact of swelling on 3D printed PEGDA hydrogels remains an area that requires further investigation. While studies have mainly focused on water and PBS, little exploration has been carried out on the effects of cell culture media on these hydrogels. Understanding their behavior and properties in a realistic cell culture setting is crucial for future research. Additionally, the long-term stability of these hydrogels is crucial, and longer-term studies spanning months or even years are required to establish their shelf life. The relationship between different environments, such as temperature and cell culture medium, and the behavior of these hydrogels also needs to be thoroughly studied, including their reversibility as a result of temperature change.

Furthermore, the stability of the hydrogels based on their storage and transportation conditions and how dehydration and rehydration may affect their long-term physical and mechanical stability requires investigation. Improving our understanding of the fundamental nanomechanical properties of 3D PEGDA hydrogels and how altering fabrication parameters such as layer thickness and UV dosage can impact their surface nanomechanical behavior uniformity is essential. Future studies should also focus on understanding the correlation between force and the nanomechanical properties of these 3D printed PEGDA hydrogels in all directions, which can significantly impact cell adhesion, differentiation, and integration with the host microstructure.

In conclusion, further research is needed to enhance the effectiveness of 3D printed PEGDA hydrogels in biomedical applications, particularly in the study of contractile tissue studies. Ensuring their physical, chemical, dimensional, and mechanical stability, as well as structural integrity, is critical for achieving positive outcomes.

**Figure 11 polymers-15-02341-f011:**
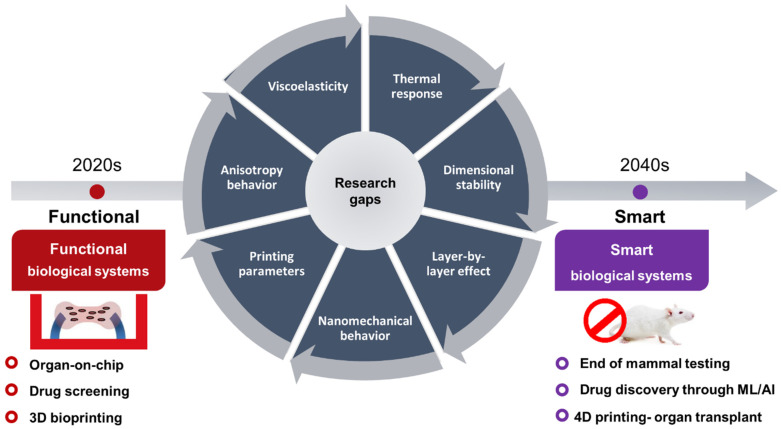
Roadmap of 3D printing technologies and their role in end of mammal testing and complete 4D organs. Showcasing the identified research gaps in 3D PEGDA hydrogels and ways to bridge the gap towards animal-free contractile tissue studies.

## 6. Conclusions

In conclusion, this review has highlighted the recent advances in additive manufacturing of PEGDA hydrogels and their potential for tissue engineering applications. The various printing techniques, such as extrusion-based, inkjet-based, and stereolithography-based printing, have been discussed, along with their advantages and limitations. The physicomechanical properties of PEGDA hydrogels and biocompatibility have been shown to be highly tunable and can be tailored to match specific tissue engineering requirements. The reviewed literature indicates that PEGDA hydrogels possess many favorable properties that make them suitable for use in tissue engineering, such as excellent biocompatibility, high water content, and the ability to support cell growth and proliferation. However, there are still many research gaps that need to be addressed in the future, such as investigating the optimal conditions for PEGDA hydrogel fabrication and functionalization, as well as the effects of different printing techniques on the resulting structures and properties. Additionally, the potential of layer-by-layer printing to improve the mechanical properties and functionality of PEGDA hydrogels should be further explored. Further in-depth in vitro and in vivo studies are needed to elucidate the mechanisms underlying the interactions between cells and PEGDA hydrogels and optimize the design of scaffolds for specific tissue engineering applications. Despite these challenges, continued exploration and advancement in the area of additive manufacturing of PEGDA hydrogels will lead to significant breakthroughs in tissue engineering, and ultimately improve the quality of life for patients in need of tissue replacement therapies. As the technology continues to improve and become more widely available, PEGDA hydrogels have the potential to become a versatile platform for various biofunctional living systems such as artificial organs, organ-on-chip, microphysiological systems, and drug testing models.

## Data Availability

Not applicable.

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
