# Peer review of "Additive Manufacturing and Physicomechanical Characteristics of PEGDA Hydrogels: Recent Advances and Perspective for Tissue Engineering"

_polymers, 2023, doi:10.3390/polym15102341_

Round 1
Reviewer 1 Report
Review
The manuscript, “Additive Manufacturing and Physicomechanical Characteristics of PEGDA Hydrogels: Recent Advances and Perspective for Tissue Engineering” by Khaliliet al. reviewed recent advancements in the use of poly(ethylene glycol) diacrylate (PEGDA) hydrogels for tissue engineering applications. It also discussed the existing challenges and future prospects of engineering 3D PEGDA hydrogels for tissue engineering and organ-on-chip devices.
Comments and Suggestions for Authors:
1- Please correct the keywords according to MeSH.
2- In lines 71-72, the sentence "However, there are still critical gaps in knowledge regarding the behavior and properties of 3D printed PEGDA hydrogels" is said. Explain more about the limitations of knowledge and features of 3D printed PEGDA hydrogel.
3- In line 84, surface elastic modulus and topography are mentioned as factors influencing the application of hydrogel. Please give examples of surface elastic modulus and different topography in previous studies.
4- Please provide a table of tissue engineering studies that have used PEGDA hydrogel. Specify the type of study and the cells used in it in this table.
5- Why do you mean by "multi-material" in "Extrusion-based printing technologies are developed further to include multi-functional, multi-material 3D printing that can be utilized to print PEGDA hydrogels with fabrication strategies for living tissues and soft robotics." in lines 131-133?
This article needs grammar editing and English review. For example in line 14 “is” must changed to “are”.
Reviewer 2 Report
Aria et al. makes a mini review on the recent advancements in the use of poly(ethylene glycol) diacrylate (PEGDA) hydrogels for tissue engineering applications. This is clearly a hot subject of research worthy of a timely review. Generally, this review was well prepared with all important works on the reviewed field included. The useful information summarized in this work merits the publication in Polymers. The following minor concerns should be further addressed prior to acceptance.
1. The uniqueness of this review relative to the published reviews with a similar research topic on PEGDA-based hydrogels for biomedical applications remain unclear, which should be highlighted clearly in the abstract and introduction sections.
2. The effect of PEGDA molecular weight on the properties and performance of the hydrogels for tissue engineering applications should be discussed.
3. How about the in vivo degradation properties of the PEGDA-based hydrogels?
4. In the last section of this manuscript (Future perspectives), please add more contents about the current problems that restrict application and clinical translation of PEGDA-based hydrogels for tissue engineering applications. In addition, please also point out future direction and solution to solve these problems. This will make this review more valuable and can provide the audience more information about what could be done in the future.
Reviewer 3 Report
In thıs review, the authors summarized "Additive Manufacturing and Physicomechanical Characteristics of PEGDA Hydrogels: Recent Advances and Perspective forTissue Engineering". It is a comprehensive review however it can be organized more so it can be understood better.
The figures (5e and 11 b) could be presented more simpler.
Tables 1,2 and 3 are good, long and detailed tables but how they can be summarized? These tables show us what? I think that all these information should show and help us to understand something at the end. This must be pointed out.
Form the title additive manufacturing and the process conditions will also be mentioned in the manuscript, including rheological properties. However this part was not the part of the manuscript. The characteristics of hydrogels produced from additive manufacturing depend on additive manufacturing conditions.
The similarity report shows 50% similarity, and it was 21% without bibliography from turnitin. The similarities could be decreased.

Round 2
Reviewer 1 Report
I dont have any comment.
Reviewer 3 Report
The manuscript can be accepted.